# Unravelling Work Drive: A Comparison between Workaholism and Overcommitment

**DOI:** 10.3390/ijerph17165755

**Published:** 2020-08-09

**Authors:** Lorenzo Avanzi, Enrico Perinelli, Michela Vignoli, Nina M. Junker, Cristian Balducci

**Affiliations:** 1Department of Psychology and Cognitive Science, University of Trento, 38068 Rovereto, Italy; enrico.perinelli@unitn.it (E.P.); michela.vignoli@unitn.it (M.V.); 2Department of Social Psychology, Goethe University Frankfurt am Main, 60323 Frankfurt, Germany; junker@psych.uni-frankfurt.de; 3Department of Psychology, University of Bologna, 40127 Bologna, Italy; cristian.balducci3@unibo.it

**Keywords:** workaholism, overcommitment, burnout, personality, other-report

## Abstract

Workaholism and overcommitment are often used as interchangeable constructs describing an individual’s over-involvement toward their own job. Employees with high levels in both constructs are characterized by an excessive effort and attachment to their job, with the incapability to detach from it and negative consequences in terms of poor health and job burnout. However, few studies have simultaneously measured both constructs, and their relationships are still not clear. In this study, we try to disentangle workaholism and overcommitment by comparing them with theoretically related contextual and personal antecedents, as well as their health consequences. We conducted a nonprobability mixed mode research design on 133 employees from different organizations in Italy using both self- and other-reported measures. To test our hypothesis that workaholism and overcommitment are related yet different constructs, we used partial correlations and regression analyses. The results confirm that these two constructs are related to each other, but also outline that overcommitment (and not workaholism) is uniquely related to job burnout, so that overcommitment rather than workaholism could represent the true negative aspect of work drive. Additionally, workaholism is more related to conscientiousness than overcommitment, while overcommitment shows a stronger relationship with neuroticism than workaholism. The theoretical implications are discussed.

## 1. Introduction

Workaholism describes the behavioral characteristics of workers who tend to work excessively and sacrifice their personal life in favor of extra-involvement in work [1]. Hence, for workaholics, their work is a sort of addiction. Indeed, these workers spend a lot of time in their job, usually much more than their colleagues, and they have difficulties to detach from it after regular work hours [2]. Therefore, recovery is impaired and this could lead to health-related problems and job burnout [3]. As such, workaholism shows a moderate positive relation with burnout and negative relation with psychological and mental health and life satisfaction [4]. Most of the authors who have studied workaholism define it as an addiction to work, and, as Clark et al. outline, addiction “involves compulsion and preoccupation with the behavior, loss of self-control, and continued engagement in the behavior despite negative consequences” [4] (p. 1838). Workaholics tend to commit themselves to extra duties, work long hours, and “keep many irons in the fire” [5]. Compulsion to work and working in excess represent the most accepted and consensual dimensions of workaholism recoverable in the literature, but other dimensions, such as work enjoyment, have been postulated to be part of the construct by some authors [4]. In fact, although this construct has been deeply studied, “one of the main issues hindering theoretical and empirical progress regarding the study of workaholism is a lack of agreement on what workaholism actually is” [4] (p. 1837). In particular, its distinctiveness to similar constructs is not yet clear.

One of such similar constructs is overcommitment [6]. Overcommitment was introduced by Siegrist [7] in the Effort–Reward Imbalance model in order to consider a personal disposition able to explain why some people exert a disproportionate effort in their job even in the absence of adequate rewards. Overcommitment describes workers with a motivational pattern that leads them to work excessively, being unable to detach from their job [7]. These workers are characterized by a high need for approval and a perceptual distortion about their job demands and available coping resources. Consequently, they “may expose themselves more often to high demands at work, or they exaggerate their efforts beyond what is formally needed” [8] (p. 1485). This perceptual distortion could explain why these workers have difficulties in “accurately assessing cost–gain relations” in the workplace, leading them to be more susceptible to exhaustion and other health problems than their colleagues [9] (p. 164). Similar to what has been found for workaholism, overcommitment has shown significant relationships with burnout and poor well-being [10,11].

The main aim of our paper is to try to disentangle these two related constructs, exploring through partial correlations, the nomological network in which workaholism and overcommitment are nested. Furthermore, building on relevant theory, we propose and test a mediational model, in which overcommitment is considered as the mediator of the relationship between workaholism and burnout. 

### 1.1. Relations between Workaholism and Overcommitment

Overcommitment and workaholism are often treated as synonymous or as overlapping constructs [12]. For example, Seybold and Salomone describe workaholism as an addiction to work and define workaholics as people who “make efforts to escape their private lives through overcommitment to work” [13] (p. 5). According to Bergin and Jimmieson, the differences between these two constructs are at least thin, since both workaholics and overcommitted employees “are motivated by a strong inner drive to work” [14] (p. 140; for similar considerations, see also [15,16]). In line with these theoretical considerations, in one of the few studies that measured both constructs, Littman-Ovadia et al. [17] found a strong correlation between workaholism and overcommitment (*r* = 0.52), as well as a similar pattern of correlations between the two constructs and criterion variables, such as burnout. The authors concluded their discussion by arguing that: “it may well be that workaholism [*is*] a personal disposition similar to overcommitment” [17] (p. 16). Andreassen et al. [18] found an even stronger correlation between workaholism and overcommitment (*r* = 0.62) in their study, and again, similar correlations between both constructs and burnout. Despite their theoretical and empirical similarity, their full nomological network could be different to some extent, and further investigation may help us in understanding their overlapping and distinctive aspects. 

### 1.2. The Role of Job Demands and External Rewards

A first characteristic that differentiates workaholism and overcommitment is related to the role played by external factors, such as job demands and external rewards. For example, Snir and Harpaz [19] conceptualized workaholism as a stable investment of time and energy in the job, regardless of external demands. However, Clark and colleagues [4] found a consistent meta-analytic correlation between workaholism and role overload (*ρ* = 0.52, 95% CI: 0.458, 0.577). Currently, it is not clear whether a highly demanding workplace or the existence of an overwork climate are antecedents of workaholism [20], or whether, on the contrary, workaholics tend to increase their work activities and fuel an overwork climate at work [21]. It is also possible that workaholics tend to exaggerate the perception of work overload for a self-serving attribution bias—erroneously ascribing their extra efforts to the high demands requested by their job or supervisors [4]. Nonetheless, it could be reasonable to hypothesize that a strong link between job demands and workaholism exists. Regarding the role played by external rewards, Souckova et al. suggested that external factors could play a small or insignificant role, since workaholism can “be regarded as an internal compulsion rather than as a reaction to external incentives” [22] (p. 71). Additionally, Clark et al. [4] specify that “workaholics do not engage in excessive work due to external factors” such as, for example, pressure by their organization or supervisor to work beyond standard working time, although organizational incentive systems may act as a reinforcement for work addiction behaviors (p. 1838). Moreover, when they perceive that their motivations and aspirations at work are met, workaholics—at least the achievement-oriented type—should show high job satisfaction [23]. Indeed, employees experience great job satisfaction based on their positive evaluation of their work experience.

On the contrary, external factors are central in the overcommitment definition. Siegrist initially proposed overcommitment as a personality construct, very similar to a type A behavior pattern, even if it was “not conceptualized as a traditional personality trait” [9] (p. 164). Overcommitment represents a motivational pattern, internal to the workers that lead them to misjudge both demands and resources. Overcommitment affects how external factors are appraised, and at the same time, overcommitment “is often elicited and reinforced by external work pressure”, amplifying the detrimental effect of this motivational pattern on health and well-being [9] (p. 164). Furthermore, employees who evaluate in a positive and fair manner the rewards received in return for their efforts should experience positive emotional states, such as job satisfaction. However, overcommitted employees could be less able to adequately evaluate the rewards obtained, and overall, they could be more affected than other colleagues from the absence of rewards or by a negative evaluation of working conditions, this leading to lower levels of job satisfaction. In this sense, the perception of external factors in terms of both demands and rewards plays a crucial role in overcommitment and should be reflected by stronger relationships between overcommitment and constructs related to the working conditions and their correlates, such as job demands and job satisfaction. 

Peiperl and Jones [24] propose to differentiate workaholics from overworkers by using two independent dimensions: perceived effort and perceived return. Following Peiperl and Jones [24], workaholics are “those who work too much but feel that the rewards arising from their work are at least equitably distributed between themselves and the organizations that employ them”. On the contrary, overworkers are defined “as people who work too much (in their own terms) just as workaholics do, but at the same time feel that the returns from their work are inequitably distributed in favor of the organization” [24] (p. 374). In this sense, rewards and their perception in terms of equity could play a fundamental role in explaining the difference between workaholism and overcommitment. Workaholics should tend to perceive more equity in received rewards, and in this sense, they should experience more satisfaction with their job than overcommitted workers. Apparently, contrary to this conclusion, in their meta-analysis, Clark et al. [4] found a negative and significant correlation between workaholism and job satisfaction (*ρ* = −0.12, 95% CI: −0.224, −0.013). Interestingly, this correlation was only found in published studies, whereas the same correlation was significant and positive (*ρ* = 0.06, 95% CI: 0.011, 0.114) in unpublished studies. Additionally, Clark et al. [4] also reported a consistent and positive correlation between workaholism and work enjoyment (*ρ* = 0.28, 95% CI: 0.223, 0.348). This may suggest that the relationship between workaholism and job satisfaction could be complex and worthy of more investigation. 

Starting from these theoretical considerations and empirical evidence, suggesting that overcommitment—differently from workaholism—may be more malleable and ‘reactive’ to external stimuli and conditions, we hypothesize that overcommitment is more related to work environmental factors and their correlates, namely job demands and job satisfaction, when compared to workaholism. 

### 1.3. The Role of Personal Dispositions

Another aspect that could be useful to disentangle the peculiarities of workaholism and overcommitment is the role played by personal dispositions. Both workaholism and overcommitment have been studied in relation to personality traits and characteristics. As outlined by Patel and colleagues, “research would benefit from focusing upon the personality traits and situational characteristics that influence the manifestation of problematic symptoms that may lead to work addiction” [25] (p. 13). Two important personality traits that could be differentially related to workaholism and overcommitment are conscientiousness and neuroticism, that is, two Big Five personality factors [26]. People with high levels of conscientiousness are well-organized and achievement-oriented, and they are able to control their impulsive behaviors. In an organizational context, highly conscientious employees are characterized by competence and achievement striving. They are perseverant, rigorous, diligent, goal-oriented, and reliable workers. Neuroticism refers to people who continuously feel negative effects. Neuroticism represents an individual stable tendency to experience negative emotions, such as anger, guilt, or anxiety. Individuals with high levels of neuroticism tend to be depressed, frustrated, and worried, and they focus more selectively on negative aspects of themselves and others. Although conscientiousness and neuroticism are related, their meta-analytic correlation is moderately low (*ρ* = −0.31, IPIP questionnaire [27]). 

Achievement-related traits could have many characteristics compatible with both workaholism and overcommitment. In particular, conscientiousness could be a significant antecedent of both constructs. Nevertheless, previous research found mixed results about the correlation between conscientiousness and both workaholism and overcommitment. For example, Clark and colleagues [4] found a positive but modest correlation between conscientiousness and workaholism (*ρ* = 0.16, 95% CI: −0.007, 0.331). However, their meta-analysis included only five studies on this relation. Additionally, other studies have found a higher, positive, and significant relationship between workaholism and conscientiousness [21]. Concerning overcommitment, its correlation with conscientiousness has been found to be even more modest and negative and non-significant (*r* = −0.12, *p* > 0.05, *N* = 224) [28]. 

Based on the idea that workaholism has been considered a true obsession for work and that obsessive-compulsive personality disorder has, among its correlates, an excessive conscientiousness and scrupulousness (see [29]), we argue that workaholism shares several cognitive and behavioral aspects with conscientiousness. Indeed, a study has found that extremely conscientious individuals may show perfectionism and compulsive behavior similar to workaholics [30]. Additionally, similar to conscientiousness, workaholism is viewed as a fairly stable personal disposition [23]. On the contrary, the notion of overcommitment has been developed by Siegrist [7] as a dysfunctional coping style with the demands of the job fueled by a strong need for approval. Aspects such as compulsive behavior, scrupulousness, and perfectionism—i.e., those that can be found in both workaholism and conscientiousness—seem to be less central for overcommitted employees. Thus, it may be expected that workaholism, differently from overcommitment, shows a significant and positive relationship with conscientiousness. 

Employees high in neuroticism tend to avoid very stressful and demanding situations, but this tendency could be compensated by their need to increase self-esteem through work, leading them toward working hardly [4,22]. Clark et al. [4] reported a low correlation between workaholism and neuroticism (*ρ* = 0.06, 95% CI: −0.248, 0.378), although more recent studies found a moderate and positive significant relationship between the two (e.g., [21]). On the contrary, the correlation between overcommitment and neuroticism has been found to be higher (*r* = 0.30, *p* < 0.05, *N* = 224) [28]. This may be expected by examining the root of the overcommitment construct, which includes in its definition aspects such as impatience and disproportionate irritability [7,8]—that is, emotional states that characterize neurotic individuals. Thus, we expect that although both workaholism and overcommitment may be positively related to neuroticism, a stronger neurotic component made by pervasive negative feelings is present, especially in overcommitment.

Starting from these considerations, we hypothesized that conscientiousness would be more strongly correlated with workaholism than overcommitment. On the contrary, neuroticism would correlate more strongly with overcommitment than workaholism.

Previous research has mainly studied the relationship between workaholism, overcommitment, and personality dispositions using self-reported personality surveys. However, one issue with self-report measures is that they could lead to social desirability bias, which is an individual’s tendency to present themselves in a favorable manner. This could be especially true for personality traits. One remedy to this possible bias consists of using other-report informants. In their meta-analysis, Connelly and Ones [31] found that other-report measures showed a strong reliability and accuracy in personality traits measurement. Furthermore, other evaluations could improve criterion validity, and this is particularly true for conscientiousness and neuroticism [31]. Thus, in the present study, we will explore the relationship between workaholism and overcommitment, on the one hand, and neuroticism and conscientiousness, on the other hand, not only by using self-reported data, but also other-reports, looking at whether our hypotheses concerning their relationships will hold with both self-reported and other-reported personality data.

### 1.4. The Mediational Role of Overcommitment in the Relation between Workaholism and Burnout

Finally, it is possible that workaholism represents a dispositional factor fueling overcommitment to work. Indeed, many definitions of work addiction postulate that workaholics are too much committed to their jobs. Since workaholism represents an internal compulsion to overwork, it should be less influenced by external factors than overcommitment, which—on the contrary—is extremely related to the employees’ judgement about the external demands and rewards detectable in job environments. Furthermore, as we previously reported, both overcommitment and workaholism are related to poor health and well-being. In particular, both constructs are related to job burnout [4,13]. Work-related burnout represents the “degree of physical and psychological fatigue and exhaustion that is perceived by the person as related to his/her work” [32] (p. 197). In this sense, burnout is related to the degree to which employees attribute their fatigue and emotional exhaustion specifically to work factors. Since we proposed that overcommitment is more susceptible to external factors than workaholism, we expect that the relation between overcommitment and burnout will be higher than the relationship between workaholism and burnout. Such an idea may also be derived by a stronger relationship between overcommitment and neuroticism, which is the best predictor of emotional exhaustion—the core component of burnout [33]. Additionally, based on the idea that workaholism may act as the personality component fueling overcommitment, and that, thus, there may be a potential causal link between the two, we also propose that overcommitment may act as a mediator in the path linking workaholism to burnout. In other words, we hypothesize that the dysfunctional pattern observable in overcommitted workers that leads to burnout has a personality root made by workaholism.

## 2. Materials and Methods 

### 2.1. Participants

We collected data with two different types of survey: online and paper and pencil. We used snowball sampling for the online collection. By using this method, 128 participants were contacted, of whom 57 (44.5%) also provided the other-reported data and for these reasons were joined in our final sample. For the paper and pencil survey, we sampled workers employed in two different organizations. The first was an accountants office of 48 employees, of whom 34 (70.8%) provided both self- and other-evaluations. The second consisted of 42 out of 47 (89.36%) employees of an educational services cooperative. Overall, the final response rate was 59.6%. Data were collected in 2019 in Italy. All participants were informed about the main aim of the research project which focused on factors affecting health and well-being at work, and they had to accept the participants’ informed consent attached to the survey. Participation was voluntary. The final sample was composed of 133 workers, 55 men (41.4%) and 78 women (58.6%), of various sectors: 16 waiters (12.03%), 34 accountants (25.56%), 42 employees of an educational services cooperative (31.58%), and 41 did not report their type of occupation (30.83%). Participants’ age was distributed as follows: 61 (45.9%) were ≤30 years old; 55 (41.4%) aged between 31 and 50; 17 (12.8%) ≥ 51 years old. An anonymous code has been created in order to match self- and other-report data. Other-report surveys were filled in by spouses (17.3%), boyfriends/girlfriends (21.4%), parents/brothers/sisters (26.5%), friends (32.7%), and others (2%). 

Since we used two different modalities of data collection, online vs. paper and pencil, a series of *t*-tests on the main study variables were conducted in order to check for potential differences in the study variables as a function of the data collection modality. There were no differences in job demands, job satisfaction, overcommitment, burnout, conscientiousness (both self- and other-reported), and neuroticism other-reported, as a function of the data collection modality. Significant differences were found in self-reported neuroticism and workaholism. In particular, participants at the online survey reported higher levels of workaholism (*t*(131) = −2.27; *p* < 0.5), and higher levels of neuroticism (*t*(131) = −2.44; *p* < 0.5) than participants in the paper and pencil survey. We ran ad hoc analyses (see below) to consider the potential implications of these differences.

### 2.2. Measures

Workaholism: Workaholism was measured by means of the Italian version of the 10-item Dutch Work Addiction Scale (DUWAS) [5,34]. Item examples were “I feel that there’s something inside me that drives me to work hard” and “I stay busy and keep many irons in the fire”. Responses were given on a Likert scale varying from 1 (strongly disagree) to 5 (strongly agree).

Overcommitment: Overcommitment was measured with the six-item scale developed by Siegrist [8]. Responses were given on a Likert scale ranging from 1 (disagreed) to 4 (agreed). An item example is “I get easily overwhelmed by time pressures at work”.

Job demands: Perceived levels of job demands were measured by means of eight items measuring the job demands dimension of the Italian version of the Stress Indicator Tool [35,36]. Reponses were given on a Likert scale ranging from 1 (“never”) to 5 (“always”). Sample items: “I have to work very intensively” and “I have unrealistic time pressures”.

Burnout: Work-related burnout was measured by means of seven items constituting a subdimension with the same name of the Italian version of the Copenhagen Burnout Inventory (CBI) [13,32]. Item examples are “Does your work frustrate you?”, and “Do you feel worn out at the end of the working day?”. Responses were provided on an intensity rating scale (1 = not at all; 5 = very much) or on a frequency rating scale (1 = never; 5 = always), depending on the content of the question.

Job satisfaction: Job satisfaction was measured with a single item: “Generally speaking, I am very satisfied with my job”. Responses were given on a Likert scale ranging from 1 (to a very low degree) to 5 (to a very high degree).

Self-reported personality traits: Self-reported conscientiousness and neuroticism were measured through four items each gathered from the International Personality Item Pool–Five Factor Model (IPIP–FFM) [37]. Participants were asked how well a series of statements described them, with responses given on a Likert scale ranging from 1 (it does not describe me at all) to 5 (it describes me completely). Item examples were “Like to order” (conscientiousness) and “Get stressed easily” (neuroticism).

Other-reported personality traits: Conscientiousness and neuroticism were measured by other informants (see “Participants” section), with the same items used for the self-reported version of the investigated personality traits. In this case, however, items asked how well each statement described the target worker. Responses were given on a Likert scale ranging from 1 (it does not describe him/her at all) to 5 (it describes him/her completely).

Control variables: We controlled for age and gender, since both variables have been found to affect employee burnout [38].

### 2.3. Ethical Aspects

This research was conducted in line with the Helsinki Declaration as well as the Italian data protection law (Legislative Decree No. 196/2003). Participants did not receive any reward for their participation, and they decided to take part in this study voluntarily. Furthermore, they were made aware that they could withdraw from the research at any time. Participants received both the survey and an attached letter in which the purpose of the study and the ethical considerations were explained (e.g., anonymity and privacy, data treatment, and so on).

### 2.4. Modeling Strategy

This study is explorative in nature. All analyses were run with the statistical software SPSS version 19. First, we computed descriptive statistics (including reliability) for each study variable. Second, after deriving a single composite score for each variable (i.e., the items’ mean), we investigated the zero-order correlations of workaholism and overcommitment with the other constructs. Third, we computed partial correlations between workaholism and the other constructs (e.g., job demands), controlling for overcommitment and partial correlations between overcommitment and the other constructs, controlling for workaholism. In partial correlations, the variance that the controlled variable shares with both the correlating variables is removed [39]. More specifically, a partial correlation “is the correlation between the two sets of residuals formed from the prediction of the original variables by one or more other variables” [39] (p. 528). Hence, in this way, we investigated (a) the relationship between workaholism and other constructs, without the shared variance with overcommitment, and (b) the relationship between overcommitment and other constructs, without the shared variance with workaholism. Fourth, we ran a mediational regression model in which we hypothesized that the relationship between workaholism and burnout would be mediated by overcommitment, controlling for sociodemographic (age and gender), contextual (job demands), and personal (conscientiousness and neuroticism other-reported) variables. The bias-corrected confidence interval of the indirect effect was estimated through 5000 bootstrap resamplings. This latter analysis was run with the SPSS macro PROCESS (Model 4) [40].

## 3. Results

Table 1 reports the descriptive statistics, reliabilities (Cronbach’s alpha), zero-order correlations, and partial correlations (see Table A1 in Appendix A for the full correlation matrix). Looking at the descriptive statistics, it can be seen that all variables were normally distributed. The reliabilities were also acceptable for all scales (all values were above the recommended threshold of 0.70).

The zero-order correlations showed a strong relationship between workaholism and overcommitment (*r* = 0.55, *p* < 0.001). However, looking at their pattern of correlations with other variables, some differences deserve to be noted. For example, it can be seen that job demands, burnout, job satisfaction (in reversed direction), and neuroticism were more strongly related to overcommitment than to workaholism, whereas conscientiousness was more strongly related to workaholism than to overcommitment. This first cluster of findings may suggest that workaholism is more strongly related to a stable personal component (i.e., conscientiousness), whereas overcommitment is more strongly related to potential indicators of job maladjustment. 

Analyses of partial correlations further highlighted the difference between the two patterns of correlations. After controlling for overcommitment, the relationship between (a) workaholism and job demands decreased from 0.43 (*p* < 0.001) to 0.18 (*p* = 0.040); (b) workaholism and job satisfaction increased from 0.01 (*p* = 0.872) to 0.17 (*p* = 0.052); (c) workaholism and burnout decreased from 0.33 (*p* < 0.001) to −0.07 (*p* = 0.452); (d) workaholism and self-reported neuroticism decreased from 0.29 (*p* < 0.01) to 0.00 (*p* = 0.988); (e) workaholism and self-reported conscientiousness increased from 0.28 (*p* < 0.01) to 0.42 (*p* < 0.001). Interestingly, the two latter changes were confirmed (but with a lower magnitude) also for other-reported personality traits. Yet, after controlling for workaholism, the relationships between overcommitment and the examined variables remained quite similar to their zero-order correlations. For example, the zero-order correlation between overcommitment and burnout was 0.67 (*p* < 0.001), while the partial correlation was 0.63 (*p* < 0.001). The only exception to this pattern was the relationship between overcommitment and conscientiousness, which changed from −0.13 (*ns*) to −0.35 (*p* < 0.001) after controlling for workaholism (a similar pattern was found also for other-reported conscientiousness). Hence, these findings support the hypothesis that both workaholism and overcommitment have a unique portion of variance that covaried in a different manner with the investigated constructs. In particular, these results suggest that variables indicating problematic working conditions and outcomes are more strongly related to (the unique variance of) overcommitment, rather than workaholism.

Table 2 reports the results of the mediational analysis. In the first step (a-path), we can see that overcommitment (mediator) was significantly and positively related to workaholism (b = 0.47, *p* < 0.001). Additionally, among other variables, only job demands and neuroticism (other-reported) were significant and positive predictors of overcommitment. In step two (b-path), burnout (dependent variable) was affected by overcommitment (b = 0.61, *p* < 0.001), which was the only significant predictor (no effect was found in relation to any other variables). Finally, as we can see in the lower part of Table 2, bias-corrected confidence intervals estimated through 5000 bootstrap resamplings did not include the value of 0, thus, attesting the significance of the indirect effect (0.29; 95% CI: 0.16, 0.43) from workaholism to burnout through overcommitment. In conclusion, overcommitment fully mediated the relationship between workaholism and burnout. In order to strengthen our hypothesis, we also ran a similar analysis in which we exchanged the role of overcommitment (now the independent variable) and workaholism (now the mediator). Results attested a non-significant mediation effect (indirect effect: −0.02; 95% CI: −0.08, 0.03), supporting the hypothesis that in our data, workaholism did not mediate the relation between overcommitment and burnout. 

To prevent social desirability bias, we ran the mediation analyses only by means of other-reported measures of personality traits. However, in order to detect potential differences, we reran analyses using self-reported traits and found that the results were substantially equivalent to those found with other-reported measures.

## 4. Discussion

Work addiction represents an irrational over-involvement at work with potential detrimental effects for employees’ well-being and health [4]. The role of work addiction could become even more central in this time of work intensification, in which the current profound economic and organizational changes have enormously increased the “amount of effort employees need to invest in their work” [41] (p. 1). Despite the large body of literature concerning the topic of work addiction, many theoretical issues linked to the nature of this phenomenon remain unresolved. In this study, we tried to unravel the relationship between two constructs often quoted as types of work addiction: workaholism and overcommitment. These two constructs have been often treated as synonymous, and indeed, they are consistently positively related to each other also in our study. Additionally, they share a similar nomological network of variables. However, by using partial correlations, we showed that despite a certain extent of overlap, they are uniquely related to different organizational and personal aspects. 

Our findings could shed light on the disagreement in theoretical definitions of workaholism, as well as on some inconsistent findings in the literature. In particular, Clark and colleagues outlined that for some scholars, workaholics “greatly enjoy[*s*] the act of working” [4,42] (p. 5), whereas the absence of work enjoyment is the true characteristic of workaholics for other researchers [43]. Empirically, Clark et al. [4] found incongruent findings in the relation between workaholism and job satisfaction if differentiating between published and unpublished studies. Our results are consistent with the results of the unpublished studies, suggesting that after removing the common variance shared with overcommitment, workaholism actually shows a small, positive, even if not significant (*p* = 0.052), relation with job satisfaction.

However, this does not mean that workaholism fosters positive emotions at work. A possible explanation of these results may be the floating experience of satisfaction that characterizes workaholics. For example, Sussman [44] speculated that rather than living a durable state of satisfaction, workaholics—like individuals affected by other addictions—could have experience of fugacious moments of work enjoyment, in correspondence for example with receiving wages, promotions, or new responsibilities at work. Thus, future studies could investigate this hypothesis also using diary or longitudinal studies that allow monitoring the variables trends over time. On the contrary, and as expected, overcommitment seems to be more strongly related to external factors, in particular, job (dis-)satisfaction, than workaholism. Consistently with the role played by job satisfaction for rewards, in a longitudinal study, Avanzi et al. [45] found that overcommitment was a predictor of burnout over time, especially for dissatisfied employees. These results seem to reinforce the distinction suggested by Peiperl and Jones [24], between workaholics and overworkers. 

Another area of disagreement detectable in the literature concerns the “role of external and contextual pressures on workaholism” [4] (p. 1841). Do workaholics create or select jobs characterized by high levels of job demands or, on the contrary, it is the overwork climate of the organization that pushes employees toward an extra-commitment to work? Our findings suggest that, once the common variance shared with overcommitment is removed, workaholism may be less related to job demands, than one could expect. This seems to confirm that workaholics are mainly driven to hard work by an internal pressure rather than external factors. Instead, overcommitment, coherently with its definition, appears to be more strongly related to external pressures than workaholism. Thus, an overwork climate could act more strongly on overcommitment than on workaholism. In any case, the direction of the relationship between overcommitment and working conditions can be ascertained only by using longitudinal designs.

Furthermore, we also found differences between workaholism and overcommitment in relation to two important personality traits: conscientiousness and neuroticism. In their meta-analysis, Clark et al. [4] found weak associations between workaholism and conscientiousness and neuroticism. Our findings outline that, while workaholism seems to be more related to conscientiousness, overcommitment is particularly related to neuroticism. These findings are interesting particularly because they are confirmed also by other-reported evaluations of personality traits, which are less affected by the social desirability bias. In particular, extremely conscientious employees may show similar characteristics to workaholics, such as perfectionism and compulsive behavior [30]. These aspects, on the contrary, seem to be less important for overcommitment. Instead, employees with high levels of neuroticism tend to experience negative emotional states, such as impatience, anger, and irritability, which are characteristics of overcommitted employees [46]. Yet, overcommitment is also related to both hostility and impatience-irritability [47]. In this case, our results confirm that workaholics could be less influenced by this personality trait. Apparently, contrary to this conclusion, Falco et al. [48] found a strong correlation between workaholism and narcissism (measured by the NPI that predominantly measures the grandiose face of narcissism), a construct related to neuroticism. However, there is evidence that neuroticism is mostly related to the vulnerable face of narcissism, rather than the grandiose one [49]. Thus, other studies are necessary to explore this relationship more deeply. While we adopted a variable-centered approach to unravel the relations between work addiction and personality traits, it could be useful in future studies to use a person-centered approach. People show at the same time some levels of neuroticism and conscientiousness, as well as other personality traits. Thus, it could be interesting to evaluate how and if workaholism and overcommitment show different relation with personality profiles. 

Finally, our findings outline that overcommitment may be (at least partially) an outcome of workaholism, representing a more proximal antecedent of employees’ well-being. Taken together, these results suggest that overcommitment could play the *true* negative role in the work addiction process. Indeed, another important result of this study is that overcommitment fully mediated the relation between workaholism and work-related burnout. Workaholics and overworkers could represent two different types of addicted employees, as suggested by Peiperl and Jones [24], but it may also be possible that workaholism plays the role of an antecedent of overcommitment. Thus, we strongly recommend researchers to test in future studies the combined effect of both these two constructs.

This study has several limitations that could represent stimuli for further investigations. A first limitation concerns our data, which are not longitudinal. Future studies should test our hypothesized mediational model by using a three-wave longitudinal design measuring both workaholism and overcommitment at the same time. Another limitation concerns the convenience sample that we used. This means that our results are not generalizable to other contexts or job occupations. It could be useful in future research to collect data by using probabilistic sampling methods in order to verify if the results emerged in the present study may be replicated. Additionally, we used self-report data for many of our main variables. We suggest for future studies to use other-report or objective measures to operationalize some of the investigated variables such as job demands. This could help to further detect to what extent workaholics actively create their job demands or, on the contrary, they are affected by a self-serving bias. Future research could also use other control variables to further disentangle workaholism from overcommitment. In the present study, to maintain the anonymity of the participants, we considered only few sociodemographic variables; future studies could additionally consider income or marital status. Regarding marital status, for example, both workaholics and overcommitted are characterized by difficulties to detach from work, thus, it is highly likely that both could have problems regarding their personal life. However, in their meta-analysis, Clark et al. [4] did not find a relation between marital status and workaholism (*ρ* = 0.00, *k* = 9). Finally, we used different modalities to gather the data. While this choice could improve the results by combining the strengths and weakness of both methods, some small differences emerged in partial correlation analyses. Online surveys could reduce the social desirability concerns, and consequently, increase the reliability of measurement. However, in the future, we suggest conducting studies in larger and representative samples, by using both online and paper and pencil modalities to ascertain if the results are consistent across data collection methods. 

All in all, our results suggest that workaholism and overcommitment could represent two partially different steps in the work addiction process. In this sense, workaholism could foster the development of employees’ overcommitment to their job, with this, in turn, leading to job burnout.

## 5. Conclusions

Our study is a first attempt to disentangle two important constructs defining the drive to work: workaholism and overcommitment. While these two constructs share many aspects, they also have unique characteristics. Our findings contribute to clarify some inconsistencies in the workaholism literature, highlighting that overcommitment could represent the *true* negative aspect of the work addiction. By using partial correlations, we found that overcommitment is more related to external factors (job demands and job *dis*-satisfaction) than workaholism. Furthermore, both self- and other-reported measures of personality traits showed a differential link between two constructs. In particular, while workaholism appears to be more strongly related with conscientiousness, overcommitment was mostly correlated with neuroticism. Finally, we found that workaholism could be an antecedent of overcommitment, that, in turn, could represent the mediator variable between workaholism and job burnout. While we believe that our research may help in workaholism’s theoretical clarification, findings should be considered with caution, since our sample was small and because the data collection derived from a non-probabilistic sampling. Thus, the generalizability of these results is reduced. 

## Figures and Tables

**Table 1 ijerph-17-05755-t001:** Descriptive Statistics, Reliability, Zero-order Correlations, and Partial Correlations.

Variables	Descriptive Statistics					Zero-order Correlations		Partial Correlations	
	*M*	*SD*	*Sk*	*Ku*	α	Workaholism	Overcommitment	Workaholism ^1^	Overcommitment ^2^
1. Workaholism	3.15	0.61	0.12	0.52	0.78	1	0.55 ^***^	1	1
2. Overcommitment	2.57	0.80	0.12	–0.43	0.84	0.55 ^***^	1	1	1
3. Job Demands	2.47	0.55	0.36	0.60	0.78	0.43 ^***^	0.56 ^***^	0.18 ^*^	0.43 ^***^
4. Burnout	2.19	0.73	0.39	0.06	0.89	0.33 ^***^	0.67 ^***^	–0.07 ^n.s.^	0.63 ^***^
5. Job satisfaction	4.06	0.85	–0.87	0.79	-	0.01 ^n.s.^	–0.23 ^**^	0.17 ^+^	–0.28 ^**^
6. Neuroticism	2.34	0.80	0.26	–0.45	0.81	0.29 ^**^	0.53 ^***^	0.00 ^n.s.^	0.46 ^***^
7. Conscientiousness	3.98	0.64	–0.26	–0.13	0.70	0.28 ^**^	–0.13 ^n.s.^	0.42 ^***^	–0.35 ^***^
8. Neuroticism (other reported)	2.47	0.86	0.20	–0.61	0.79	0.25 ^**^	0.36 ^***^	0.06 ^n.s.^	0.28 ^**^
9. Conscientiousness (other reported)	3.89	0.84	–0.69	0.29	0.81	0.19 ^*^	0.01 ^n.s.^	0.23 ^*^	–0.12 ^n.s.^

Note. *N* = 133. *M*—mean; *SD*—standard deviation; *Sk*—skewness; *Ku*—kurtosis; *α*—Cronbach’s alpha. ^n.s.^*p* > 0.10. ^+^
*p* < 0.10. ^*^
*p* < 0.05. ^**^
*p* < 0.01. ^***^
*p* < 0.001. For job satisfaction, Cronbach’s alpha could not be computed because it was measured by a single item. ^1^ Controlling for the effect of overcommitment. ^2^ Controlling for the effect of workaholism. Since we found some differences on the basis of the data collection modality, we ran our partial correlation analyses, also splitting our full sample into two subsamples (1 = paper and pencil, *N* = 73; 2 = online, *N* = 57). The main results remained substantially unchanged.

**Table 2 ijerph-17-05755-t002:** Results of Mediation Analysis.

Variables	Overcommitment (*M*) *R*^2^ = 0.47 ^***^		Burnout (*Y*) *R*^2^ = 0.47 ^***^
	*b coefficient (SE)*		*b coefficient (SE)*
Gender	0.02 (0.11)		0.08 (0.10)
Age	–0.01 (0.08)		–0.11 (0.07)
Job Demands	0.53 (0.11) ^***^		0.10 (0.11)
Neuroticism (other-reported)	0.18 (0.06) ^**^		–0.00 (0.06)
Conscientiousness (other-reported)	–0.09 (0.07)		–0.03 (0.06)
Workaholism	0.47 (0.10) ^***^		–0.06 (0.10)
Overcommitment	-		0.61 (0.08) ^***^
Indirect effect of Workaholism (*X*) on Burnout (*Y*) through Overcommitment (*M*)			
	95% Lower Limit	Estimate	95% Upper Limit
	0.16	0.29	0.43

*Note. M*—mediator; *Y*—dependent variable; *X*—independent variable; *SE*—standard error; *R*^2^—explained variance. ^**^*p* < 0.01. ^***^
*p* < 0.001. Since we found some differences on the basis of data collection mode, we ran an additional mediational analysis, also controlling for mode (1 = paper and pencil; 0 = online). Results remained unchanged.

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
