# Peer review of "Unravelling Work Drive: A Comparison between Workaholism and Overcommitment"

_ijerph, 2020, doi:10.3390/ijerph17165755_

Round 1

Reviewer 1 Report

This is a very interessant paper that aims to better understand the commonalities and differences between workaholism and overcommitment.

Introduction section

The authors evoke an interesting hypothesis about the relationship between these two concepts and the notion of reward (without naming it, they thus evoke the notion of distributive justice). However, this aspect is no longer mentioned at all in their procedure. To complete their text, the authors could therefore better explain the link they consider between job demand (which they measured) and reward or explain why they did not retain this latter point in their measurements and only focused on job demand.

Furthermore, the transitions in lines 86 to 90 could be better organized.

Results section

it is unfortunate that the authors do not describe the self-reported data at all.
They could also comment the lack of difference between their two types of measurements. What could it mean in the present context ? 

Author Response

Rev#1 This is a very interessant paper that aims to better understand the commonalities and differences between workaholism and overcommitment.

Our response: Many thanks to Reviewer#1 for his/her positive evaluation of our work.

Rev#1_Q1 Introduction section

The authors evoke an interesting hypothesis about the relationship between these two concepts and the notion of reward (without naming it, they thus evoke the notion of distributive justice). However, this aspect is no longer mentioned at all in their procedure. To complete their text, the authors could therefore better explain the link they consider between job demand (which they measured) and reward or explain why they did not retain this latter point in their measurements and only focused on job demand.

Our response: You are right. We directly measured job demands, but we did not directly measure rewards. We considered that when employees meet the reward expected, that is when employees evaluate that they received adequate rewards in return to their efforts, they should experience more job satisfaction. In other words, employees who believe that they receive fair and adequate rewards from their work will experience positive emotional states associated with the perception of self-achievement, accomplishment, and growth. Job satisfaction represents a pleasant emotional state associated with a positive evaluation of the work experience. Thus, we expect that there is a relation between rewards and job satisfaction. Consistently, for example, Kinman and Jones (2008) found in a large sample of employees a moderate correlation between rewards and job satisfaction (r = .43, N = 1,108).

Additionally, we didn’t focus directly on rewards because we didn’t find any evidence on the relationship between workaholism and rewards (see Clark et al., 2016). On the contrary, there is evidence on workaholism and job satisfaction as well as overcommitment and job satisfaction; so we decided to use job satisfaction as an indirect measure (or indication) of the rewards received by the organization. We added in the introduction the following sentence:

“Furthermore, employees who evaluate in a positive and fair manner the rewards received in return for their efforts should experience positive emotional states, such as job satisfaction.”

Kinman, G., & Jones, F. (2008). Effort-Reward Imbalance and Overcommitment: Predicting Strain in Academic Employees in the United Kingdom. International Journal of Stress Management 15(4):381-395.

Rev#1_Q2 Furthermore, the transitions in lines 86 to 90 could be better organized.

Our response: Now, we added a couple of sentences to strengthen our rationale:

“Also Clark et al. [4] specify that “workaholics do not engage in excessive work due to external factors” such as, for example, pressure by their organization or supervisor to work beyond working time, although organizational incentive systems may act as a reinforcement for work addiction behaviors (p. 1838). When they perceive that their motivations and aspirations at work are met, workaholics – at least the achievement-oriented type – should show high job satisfaction [23]. Indeed, employees experience more job satisfaction based on their positive evaluation of their work experience.”

Rev#1_Q3 Results section

it is unfortunate that the authors do not describe the self-reported data at all.

Our response:
You are right that the methods section needed more information about the sample and how we collected the data. Now we added some information:

“We collected data with two different types of survey: Online and paper and pencil. We used a snowball sampling for the online collection. By using this method 128 participants were contacted, of whom 57 (44.5%) also provided the other-reported data and for this reasons were joined in our finale sample. For the paper and pencil survey, we sampled workers employed in two different organizations. The first was an accountants office of 48 employees, of whom 34 (70.8%) provided both self- and other-evaluations. The second consisted of 42 out of 47 (89.36%) employees of an educational services cooperative. Overall, the final response rate was 59.6%. Data were collected in 2019 in Italy. All participants were informed about the main aim of the research project which focused on factors affecting health and well-being at work, and they have to accept the participants' informed consent attached to the survey. Participation was voluntary.” 

Rev#1_Q4
They could also comment the lack of difference between their two types of measurements. What could it mean in the present context? 

Our response: Actually, we found some differences between self- and other- evaluations in relations with other outcomes. For example, as it can be seen in Table 1 (Appendix A), the correlation between burnout and conscientiousness (self-reported) is -.23, whereas the correlation between burnout and conscientiousness (other-reported) is -.03. A similar difference has been found with the relations between burnout and neuroticism (.50, and .25, for self- and other-reported measures, respectively). Similar differences have been found also for the correlations between job satisfaction and both self- and other measures of conscientiousness (.25 vs .04), and neuroticism (-.41 vs -.17).

Also about our main variables, we found differences in self- and other-evaluations:

                                               Workaholism                          Overcommitment

Neuroticism self                                 .29                                          .53
Neuroticism other                              .25                                          .36
Conscientiousness self                       .28                                         -.13
Conscientiousness other                    .19                                           .01      

As we can see, only the correlation between workaholism and neuroticism seems to be similar in terms of size in both self- and other-reported measures (.29 vs.25). In the other cases, differences in size have been found: .29 vs.19; .53 vs.36, and -.13 vs .00.
We found a similar pattern also considering the partial correlations:

Workaholism                          Overcommitment

Neuroticism self                                 .00                                          .46
Neuroticism other                              .06                                          .28
Conscientiousness self                       .42                                         -.35
Conscientiousness other                    .23                                         -.12

Again, except for the partial correlation between workaholism and neuroticism, in which we found little difference in coefficients’ size in self- and other evaluations (.00 vs .06), differences in size were found for all other cases (.42 vs .23; .46 vs .28, and -.35 vs -.12).

So, while the direction of the correlation among variables is not different when it is considered the two sources of information (self- and other-reported measures), differences in size have been found. We think that these findings are consistent with our expectations.

Reviewer 2 Report

Thank you for the opportunity to review the manuscript “Unravelling work drive: A comparison between workaholism and overcommitment” for International Journal of Environmental Research and Public Health. This manuscript, reported to use cross sectional data, aims to disentangle workaholism and overcommitment by assessing their relationship (1) relevant antecedents, and (2) health consequences.  This was an ambitious manuscript, and I enjoyed reviewing it. There is much to like with this paper. Overall, it was thought provoking and enjoyable read. Importantly, this research has the potential to help remedy some of the conflicting findings in workaholism research and helps to parse out which characteristics may be more closely related to distress and burnout. Overall, it was clearly written and thought provoking.  I do, however, have some comments on how the manuscript could be improved.

Major concerns:

  • The part of the paper that introduced conscientiousness and neuroticism (lines 128-167) brought up a few questions for me. It seems likely that individuals that are “conscientious” (i.e. perfectionistic, scrupulous) would also be “neurotic” in some ways as well (i.e. anxious, focused on negative aspects of self). These concepts are different, but it seems like they are closely related. Is it likely that people would just identify with one of the concepts? If not, then does this serve as a useful differentiating factor between workaholics and over-committers? (I feel like the differences in these concepts are better explained throughout the rest of the paper, so maybe add in better descriptions of the differences up front (i.e. stable vs. unstable personality traits, maybe talk about more about how the “Big Five” personality researchers distinguish these attributes).

  • Little information is provided regarding the data collection efforts. Consequently, due to this dearth of information, I find the results to be dubious at best. Where were these data gathered? When were these data collected? What did the recruitment process look like? What was the response rate for the survey? In the abstract and when discussing study limitations, the authors state that this was a cross-sectional study. However, the authors do not provide any information on how the 133 workers actually represent a cross section of all workers in whatever location the survey was conducted. My suspicion is that the authors relied on nonprobability methods to gather these data. This is not a small issue I am raising. If my suspicion is correct, then the findings cannot be generalized beyond the sample. There is nothing inherently wrong with using a nonprobability approach, especially considering this is an exploratory study (which, by the way, should be stated in the methods section). In my view the scholarship of this work would be strengthened considerably if the author(s) would be more explicit in their description of the data collection efforts.

  • Related to the above point: if my suspicion is correct that these data are not representative of a cross section of all workers in [name of study setting], the authors need to edit text the “Results,” “Discussion,” and “Conclusion” sections to reflect an inability to generalize beyond the sample.

  • From what little information was provided regarding data collection, it is clear that the authors employed a mixed mode data collection strategy (e.g., use of both an online and paper questionnaire). What is the distribution of data across these different modes? Furthermore, the authors neglect to explain how they controlled for potential response bias related to the type of survey filled out by the respondents.

  • To my eye, there other control variables missing. For example, why was the type of job or income not controlled for? What about marital status and other household and personal characteristics? Certainly, these, for example, are important considerations if one truly desires to disentangle workaholism and overcommitment.

  • The concluding statement in the discussion section (lines 402-405) implies that the authors are summarizing the main findings of the study as it leads in with, “All in all, our results suggest…” However, the crux of this statement, “workaholism could foster the development of employees’ overcommitment to their job, and this in turn could lead to health-related problems”, is not what the mediation analysis suggests. The model measures the mediation effect of overcommitment on burnout. So, in this concluding statement, I would change “health-related problems”, which–to my understanding–aren’t included in the model, to “burnout”. I believe health is also mentioned in the abstract several times, but I was not seeing the connection to health outcomes in the analysis.

Tangential concerns:

  • In the last sentence of the introduction, it might be helpful to include that the study will also look at potential mediating relationships along with partial correlations, since the mediating relationship is one of the most important findings of this research.

Overall, I thought the article was very convincing. I certainly came away feeling like separating these concepts in future research would be the best approach. This paper has to potential to help remedy some of the conflicting findings in workaholism research and helps to parse out which characteristics may be more closely related to distress and burnout.

Author Response

Rev#2 Thank you for the opportunity to review the manuscript “Unravelling work drive: A comparison between workaholism and overcommitment” for International Journal of Environmental Research and Public Health. This manuscript, reported to use cross sectional data, aims to disentangle workaholism and overcommitment by assessing their relationship (1) relevant antecedents, and (2) health consequences.  This was an ambitious manuscript, and I enjoyed reviewing it. There is much to like with this paper. Overall, it was thought provoking and enjoyable read. Importantly, this research has the potential to help remedy some of the conflicting findings in workaholism research and helps to parse out which characteristics may be more closely related to distress and burnout. Overall, it was clearly written and thought provoking.  I do, however, have some comments on how the manuscript could be improved.

Our response: Many thanks to Reviewer#2 for his/her positive evaluation of our work.

Major concerns:

  • Rev#2_Q1 The part of the paper that introduced conscientiousness and neuroticism (lines 128-167) brought up a few questions for me. It seems likely that individuals that are “conscientious” (i.e. perfectionistic, scrupulous) would also be “neurotic” in some ways as well (i.e. anxious, focused on negative aspects of self). These concepts are different, but it seems like they are closely related. Is it likely that people would just identify with one of the concepts? If not, then does this serve as a useful differentiating factor between workaholics and over-committers? (I feel like the differences in these concepts are better explained throughout the rest of the paper, so maybe add in better descriptions of the differences up front (i.e. stable vs. unstable personality traits, maybe talk about more about how the “Big Five” personality researchers distinguish these attributes).

Our response: Thank you for your suggestion. Now, in the revised paper, we explain the nature of these two personality traits (see below). The meta-analytic correlation between these two constructs greatly varies depending on the instrument used in research (see Table 3 of meta-analysis by van der Linden, Jijenhuis, & Bakker, 2010). However, by using IPIP measures (the same that we used in our study) the meta-analytic correlation between conscientiousness and neuroticism is not very high: ρ = -.31. Thus, we think that at least partially these two constructs reflect different dimensions of human personality. Anyway, you are right that it could be useful to control for personality’s cluster. We outlined this in the discussion.

“Two important personality traits that could be differentially related to workaholism and overcommitment are conscientiousness and neuroticism, that is, two Big Five personality factors [26]. People with high levels of conscientiousness, are well-organized and achievement-oriented, and they are able to control their impulsive behaviors. At work context, highly conscientious employees are characterized by competence and achievement striving. They are perseverant, rigorous, diligent, goal-oriented, and reliable workers. Neuroticism refers to people who continuously experiment negative affects. Nuroticism represents an individual stable tendency to experience negative emotions, such as anger, guilt or anxiety. Individuals with high levels of neuroticism tend to be depressed, frustrated, and worried, and they focus more selectively on negative aspects of self and others. Although conscientiousness and neuroticism are related, their meta-analytic correlation is moderately low (ρ = -.31, IPIP questionnaire [27]).

Discussion:
“While we adopted a variable-centered approach to unravel the relations between work addiction and personality traits, it could be useful in future studies to use a person-centered approach. People show at the same time some levels of neuroticism and conscientiousness, as well as other personality traits. Thus, it could be interesting to evaluate how and if workaholism and overcommitment show different relation with personality’s profiles.

References: Van der Linden, D.; Te Nijenhuis, J.; Bakker, A. B. The General Factor of Personality: A meta-analysis of Big Five intercorrelations and a criterion-related validity study. J Res Pers 2010, 44, 315-327.

  • Rev#2_Q2 Little information is provided regarding the data collection efforts. Consequently, due to this dearth of information, I find the results to be dubious at best. Where were these data gathered? When were these data collected? What did the recruitment process look like? What was the response rate for the survey? In the abstract and when discussing study limitations, the authors state that this was a cross-sectional study. However, the authors do not provide any information on how the 133 workers actually represent a cross section of all workers in whatever location the survey was conducted. My suspicion is that the authors relied on nonprobability methods to gather these data. This is not a small issue I am raising. If my suspicion is correct, then the findings cannot be generalized beyond the sample. There is nothing inherently wrong with using a nonprobability approach, especially considering this is an exploratory study (which, by the way, should be stated in the methods section). In my view the scholarship of this work would be strengthened considerably if the author(s) would be more explicit in their description of the data collection efforts.

Our response: Yes, you are right. We used a nonprobability method to obtain our data, and now we added more information about the data collection. We also specified this in the limitation section and added a sentence in the method section on the explorative nature of our study.

“We collected data with two different types of survey: Online and paper and pencil. We used a snowball sampling for the online collection. By using this method 128 participants were contacted, of whom 57 (44.5%) also provided the other-reported data and for this reasons were joined in our finale sample. For the paper and pencil survey, we sampled workers employed in two different organizations. The first was an accountants office of 48 employees, of whom 34 (70.8%) provided both self- and other-evaluations. The second consisted of 42 out of 47 (89.36%) employees of an educational services cooperative. Overall, the final response rate was 59.6%. Data were collected in 2019 in Italy. All participants were informed about the main aim of the research project which focused on factors affecting health and well-being at work, and they have to accept the participants' informed consent attached to the survey. Participation was voluntary.

In the limitations, we added:

“Another limitation concerns the convenience sample that we used. This means that our results are not generalizable to other contexts or job occupations. It could be useful in future research to collect data by using probabilistic sampling methods in order to verify if the results emerged in the present study may be replicated. Additionally, we used self-report data for many of our main variables”.

  • Rev#2_Q3 Related to the above point: if my suspicion is correct that these data are not representative of a cross section of all workers in [name of study setting], the authors need to edit text the “Results,” “Discussion,” and “Conclusion” sections to reflect an inability to generalize beyond the sample.

Our response: We agree with the reviewer that this point need to be addresses in the paper. Thus, we added two sentences in order to do that. One is previous quoted in the discussion section, the second one is the following and now it is inserted in the Conclusion section:

“While we believe that our research may help in workaholism’s theoretical clarification, findings should be considered with caution since our sample was small and because the data collection derives from a non-probabilistic sampling. Thus, the generalizability of these results is reduced.”

  • Rev#2_Q4 From what little information was provided regarding data collection, it is clear that the authors employed a mixed mode data collection strategy (e.g., use of both an online and paper questionnaire). What is the distribution of data across these different modes? Furthermore, the authors neglect to explain how they controlled for potential response bias related to the type of survey filled out by the respondents.

Our response: Yes, as we now specified in the sample description, we used two different modalities to collect the data: online and paper and pencil survey. Following your suggestion, we ran a series of T-test, in order to ascertain if our main variables were influenced by the mode of data collection. We compared the two groups (online vs paper-and-pencil) with all our variables, and no differences emerged about job demands, job satisfaction, overcommitment, burnout, conscientiousness (both self- and other-reported), and neuroticism other-reported. Instead, some differences were found about neuroticism self-reported and workaholism. In particular, the participants in the online survey reported higher levels of workaholism (t(131) = -2.27; p < 0.5), and higher levels of neuroticism (t(131) = -2.44; p < 0.5) than participants in the paper-and-pencil survey.
We used a dummy variable (0 = online, 1 = paper and pencil) in our mediational model to account for the modality of data collection, and results remained unchanged. This new variable was positively and significantly related with overcommitment (mediator model: b = .26, p = .027), while it had no significant relation with burnout (dependent model: b = -.18, p = .101). We added a sentence and a footnote reporting these additional results.
We also re-run partial correlations by splitting our sample in two sub-samples: paper and pencil (N = 73), and online (N = 57). Results are shown in Table 1 – Revision. As we it can be seen, the partial correlations of overcommitment (controlled for workaholism) with all other variables in both samples did not show relevant changes when compared to the results in the full sample. For example, the partial correlations between overcommitment and job demands range from .38 to .46; the partial correlations between overcommitment and job satisfaction range from -.28 to -.33. The only exception is for conscientiousness other-reported. In this case, the correlation between overcommitment and this variable range from -.29, p = .03 (online survey), to -.12, p >.05 (full sample), to .01, p >.05 (paper and pencil survey).

Concerning workaholism (controlled for overcommitment), its partial correlations with some variables remained substantially unchanged across the different samples. For example, the partial correlation between workaholism and job satisfaction range from .17 to .24; the partial correlation between workaholism and conscientiousness other-reported range from .21 to .27. Some difference emerged in relation to other variables. For example, the partial correlation between workaholism and job burnout range from -.02, p >.05 (paper and pencil survey), to -.07, p >.05 (full sample), to -.20, p >.05 (online survey). Again, the partial correlation between workaholism and neuroticism self-reported range from -.17, p >.05 (online survey), to 0, p >.05 (full sample), to .04, p >.05 (paper and pencil survey). Nonetheless, in both cases, despite the different size of the coefficients, no partial correlation was statistically significant. Anyway, we report about this in the results and limitations sections.

Results:
“Since we found some differences on the basis of the data collection modality, we ran our partial correlation analyses, also splitting our full sample in two sub-samples (1 = paper and pencil, N = 73; 2 = online, N = 57). The main results remained substantially unchanged.”

Limitations:
“Finally, we used different modalities to gather the data. While this choice could improve the results by combining the strengths and weakness of both methods, some small differences emerged in partial correlation analyses. Online surveys could reduce the social desirability concerns, and consequently, increasing the reliability of measurement. However, in the future we suggest to conduct studies in larger and representative samples, by using both online and paper and pencil modalities to ascertain if the results are consistent across the data collection methods.”

Table 1 – Revision.

Variables

Zero-order Correlations

N = 133

Partial Correlations

N = 133

Partial Correlations

N = 73 (paper and pencil)

Partial Correlations

N = 57 (online)

Workaholism

Overcommitment

Workaholism1

Overcommitment2

Workaholism1

Overcommitment2

Workaholism1

Overcommitment2

1. Workaholism

1

.55***

1

1

1

1

1

1

2. Overcommitment

.55***

1

1

1

1

1

1

1

3. Job Demands

.43***

.56***

.18*

.43***

.11n.s.

.46***

.26*

.38***

4. Burnout

.33***

.67***

-.07n.s.

.63***

-.02n.s.

.53***

-.20n.s.

.74***

5. Job satisfaction

.01n.s.

-.23**

.17+

-.28**

.17n.s.

-.28**

.24(p = .07)

-.33**

6. Neuroticism

.29**

.53***

.00n.s.

.46***

.04n.s.

.53***

-.17n.s.

.49***

7. Conscientiousness

.28**

-.13n.s.

.42***

-.35***

.46***

-.28**

.38***

-.49***

8. Neuroticism

(other reported)

.25**

.36***

.06n.s.

.28**

.10n.s.

.27**

-.07n.s.

.32**

9. Conscientiousness (other reported)

.19*

.01n.s.

.23*

-.12n.s.

.27**

.01n.s.

.21n.s.

-.29*

  • Rev#2_Q5 To my eye, there other control variables missing. For example, why was the type of job or income not controlled for? What about marital status and other household and personal characteristics? Certainly, these, for example, are important considerations if one truly desires to disentangle workaholism and overcommitment.

Our response: You are right that other control variables could be used. However, to obtain anonymous surveys, we decided to collect only few socio-demographic data. This was particularly important as some participants came from two small organizations (N = 47 and N = 48) so that any additional socio-demographic information would have increased the possibility of identifying individual responses. Anyway, in Clark et al’s (2016) meta-analysis, the correlation between educational level (ρ = -.02, 95% CI: -.124, .076), marital status (ρ = .00, 95% CI: -.055, .053), and number of children (ρ = .05, 95% CI: -.062, .169) with workaholism are small and inconsistent. Only the correlation between workaholism and income (salary) appears in this meta-analysis marginally important (ρ = .13, 95% CI: -.046, .304), even if inconsistent. But, of course, future studies should explore more deeply the differences between workaholism and overcommitment in relation to other socio-demographic variables. About type of job, as specified in sample description, 41 participants did not report their type of occupation, thus we would be forced to not consider them in our analyses, further reducing our final sample (from 133 to 92). In a very coarse way and only for exploratory purpose, we built a dummy variable in which 1 represents the 42 employees of the educational services cooperative, and 0 all other participants. We inserted this dummy variable (“Type of job”) in our final mediational model, and the results did not change. Type of job had no effect in our mediator model (overcommitment: b = .06, p = .609) as well as in the dependent model (burnout: b = .02, p = .894). Thus, we believe, that at least in our sample, the type of job does not represent a relevant control variable able to substantially change the results.
We added the following sentences in the limitation section:

“Future research could also use other control variables to further disentangle workaholism from overcommtiment. In the present study, to maintain the anonymity of the participants, we considered only few socio-demographic variables: future studies could additionally consider income or marital status. Regarding marital status, for example, both workaholics and overcommitted are characterized by difficulties to detach from work, thus it is highly likely that both could have problems regarding their personal life. However, in their meta-analysis, Clark et al. [4] didn’t find a relation between marital status and workaholism (ρ = .00, k = 9).”

  • Rev#2_Q6 The concluding statement in the discussion section (lines 402-405) implies that the authors are summarizing the main findings of the study as it leads in with, “All in all, our results suggest…” However, the crux of this statement, “workaholism could foster the development of employees’ overcommitment to their job, and this in turn could lead to health-related problems”, is not what the mediation analysis suggests. The model measures the mediation effect of overcommitment on burnout. So, in this concluding statement, I would change “health-related problems”, which–to my understanding–aren’t included in the model, to “burnout”. I believe health is also mentioned in the abstract several times, but I was not seeing the connection to health outcomes in the analysis.

Our response: Thank you. Now we changed in both concluding statement and in the abstract “health-related problems” with “job burnout”.

Tangential concerns:

  • Rev#2_Q7 In the last sentence of the introduction, it might be helpful to include that the study will also look at potential mediating relationships along with partial correlations, since the mediating relationship is one of the most important findings of this research.

Our response: Thank you, now we added the following sentence in the introduction:

“Furthermore, building on relevant theory, we propose and test a mediational model, in which overcommitment is considered as the mediator of the relationship between workaholism and burnout.”

Rev#2_Q8 Overall, I thought the article was very convincing. I certainly came away feeling like separating these concepts in future research would be the best approach. This paper has to potential to help remedy some of the conflicting findings in workaholism research and helps to parse out which characteristics may be more closely related to distress and burnout.

Our response: Many thanks to Reviewer#2 for his/her positive evaluation of our work. We believe that our paper is strongly strengthened thanks to his/her suggestions.

Round 2

Reviewer 2 Report

Thank you for the opportunity to review the revised manuscript “Unravelling work drive: A comparison between workaholism and overcommitment” for International Journal of Environmental Research and Public Health.

My only remaining concern is that, throughout the manuscript, the authors persist in referring to data used in their analysis as “cross-sectional.” This is factually incorrect and should be edited accordingly. Beyond that, the authors have, in an honest and professional way addressed all my remaining concerns. Thank you.

The authors are correct in their assertion that cross-sectional data are
typically collected at one point in time. (Although it is not uncommon
for researchers to conduct a repeated cross-sectional study—often
referred to as trend study—where data are collected at two or more
points in time from different samples of the same target population).
However, what the authors describe in the paper is a nonprobability
mixed mode research design (e.g.., they used multiple modalities of data
collection: -1- online, and -2- paper and pencil) which was not
cross-sectional with regards to sampling procedures. For a study to be
correctly described cross-sectional the design must be such that a
representative sample is drawn from a target population. That is, you
take a cross-section—a slice that cuts across an entire population—and
use that to see all the different parts, or sections, of that
population. Imagine cutting out a slice of a tree trunk, from bark to
core. In looking at this cross-section, one can see all the different
parts, including the rings of the tree. In social research, for example,
you might do a cross-sectional study of a college’s student body, with a
sample that included proportional representation of freshman through
seniors. This “slice” of the population, taken at a single point in
time, allows one to make inferences about the student body as a whole,
while also allowing for cross comparisons of the different groups. In
medical research, for example, a researcher might collect
cross-sectional data from a population to investigate past smoking
habits and a current diagnoses of SARS-CoV-2. While such a study design
cannot demonstrate cause and effect, it can provide a quick look at
correlations that may exist at a particular point in time.

To reiterate, an essential and defining characteristic of a
cross-sectional study is that it is a representative sample drawn across
a target population. Because the authors relied on nonprobability
sampling approaches for data collection, these data are not
generalizable and, therefore, cannot be representative of a cross
section of their target population. Moreover, the authors did not
justify that these data are cross-sectional by providing descriptive
information comparing their sample and the stated target population
(i.e., employees from different organizations in Italy). Likewise, the
authors do not justify how such a small sample (n=133) of employees can
be representative of a cross section of employees from all of the
different organizations in Italy. However, to be clear, I do not
consider the authors’ research design to be a weakness of the study.
After all, the stated goal of this research is not to make inferences
about the target population but rather to improve our understanding of
two concepts: overcommitment and workaholism. In fact, I applaud the
author’s intention to improve (1) our understanding of and (2) analytic
assessment of overcommitment and workaholism. After all, assessment is
mediated by the precision of our concepts. As such, precision is both
operational (a matter of method) and definitional (a matter of theory).
Thus, this paper has the potential to make an important continuation.
Nevertheless, I reiterate my concern with regards to a lack of
definitional clarity in describing data used in this study. To that end,
in my reading, this paper will be strengthened by removing two
definitional inconsistencies (see lines 18-19 and 444-445).

Recommended edits:
• Lines 18-19: “We conducted a survey of employees (n=133) from
different organizations in Italy.”
• Lines 444-445: “A first limitation concerns the nature of our data.”

Author Response

Dear referee, 

thank you for your suggestion and explanation about the cross-sectional design. Now we consitently changed the two sentences suggested. 

In attachment the final version. 
